# Establishing a framework towards monitoring HCV microelimination among men who have sex with men living with HIV in Germany: A modeling analysis

Lara K. Marquez[1]☯*, Patrick Ingiliz[2,3]☯*, Christoph Boesecke[4], Ivanka Krznaric[2], Knud Schewe[5], Thomas Lutz[6], Stefan Mauss[7], Stefan Christensen[8,9], Jürgen K. Rockstroh[4], Sonia Jain[10], Feng He[10], Joel O. Wertheim[1], Natasha K. Martin[1]

1 Division of Infectious Diseases and Global Public Health, University of California San Diego, La Jolla, CA, United States of America, 2 Center for Infectiology, Berlin, Germany, 3 Hepatology Department, Henri-Mondor Hospital, INSERM U955, Créteil, France, 4 Department of Medicine, University of Bonn, Bonn, Germany, 5 ICH-Studycenter, Hamburg, Germany, 6 Infektiologikum, Frankfurt, Germany, 7 Center for HIV and Hepatogastroenterology, Duesseldorf, Germany, 8 CIM Münster, Münster, Germany, 9 Department of Gastroenterology and Hepatology, Muenster University Hospital, Muenster, Germany, 10 Herbert Wertheim School of Public Health and Human Longevity Science, University of California San Diego, La Jolla, CA, United States of America

☯ These authors contributed equally to this work.
* lkusnezo@health.ucsd.edu (LKM); p_ingiliz@web.de (PI)

**Data Availability Statement:** All relevant data are within the article and its Supporting Information files.

## Abstract

### Background

Prior to direct-acting antivirals (DAAs), HCV incidence rose among men who have sex with men (MSM) living with HIV infection in Germany despite high hepatitis C virus (HCV) treatment rates. We establish a HCV elimination modeling framework to evaluate whether existing treatment rates can achieve the World Health Organization (WHO) incidence target among MSM living with HIV in Germany.

### Methods

To evaluate progress towards HCV elimination in Germany, we adapted a previously published HCV transmission model among MSM living with diagnosed HIV. We modelled HCV incidence and prevalence until 2030 (relative to 2015) under existing treatment and DAA scale-up and explored potential impacts of disruptions in treatment and behavioral risk reduction due to the COVID-19 pandemic.

### Results

Continuing current treatment rates will result in stable HCV incidence among MSM living with HIV in Germany between 2015–2030. The WHO HCV incidence target is achievable under DAA scale-up to 100% treatment combined with treatment of those previously diagnosed and untreated (at a rate of 15%/year) and would result in greater reductions with early treatment (3 vs 6 months) reducing incidence from 4.0/100person-years to 0.8/100person-

**Funding:** This study was funded by Gilead (PI: Patrick Ingiliz, grant numbers IN-DE-987-4637, IN-FR-980-6332) (url: https://urldefense.com/v3/__ https://www.gilead.com/__;!!LLK065n_VXAQ! 23TxhSkx7A5mo3pmlA6ob408Bh5zcfkyn7lQUI_ 3fV82aNCwgCrdgBXCkgoXgwZz4ac$). LKM was supported by the National Institutes of Health (NIH) and National Institute on Drug Abuse (NIDA; https://nida.nih.gov/) Ruth L. Kirschstein Institutional National Research Service Award, T32 Postdoctoral fellowship in Substance abuse, HIV, and Related Infections (PI: Steffanie Strathdee, grant number T32 DA023356) and by the Fogarty International Center of the National Institutes of Health (url: https://www.fic.nih.gov/) and the University of California Global Health Institute under Award Number D43TW009343 (url: https:// ucghi.universityofcalifornia.edu/uc-global-health-institute). JOW acknowledges funding from the National Institutes of Health (NIH) and the National Institute of Allergy and Infectious Diseases (url: https://www.niaid.nih.gov/) under Award Number: R01 AI135992.

**Competing interests:** NKM has received unrestricted research grants and honoraria from Gilead and Merck. CB reports personal fees from AbbVie, Gilead, Johnson & Johnson, Merck Sharp & Dohme (MSD), and ViiV and grants from DZIF (German Center for Infection Research), Deutsche Leberstiftung, and Hector Stiftung. TL reports grants from AbbVie, Gilead Sciences, ViiV, Janssen-Cilag, MSD, Deutsche Leberstiftung, and DAGNÄ for scientific work. SC reports personal fees from Gilead, AbbVie, ViiV, MSD, and Indivior. SM reports personal fees from AbbVie, Gilead, and MSD. JKR reports personal fees from Abivax, AbbVie, Gilead, Janssen, Merck, and ViiV, outside the submitted work. PI has received unrestricted research grants from Gilead (IN-DE-987-4637 and IN-FR-980-6332), a research grant from Abott Laboratories, and speaker fees from Gilead, AbbVie, ViiV, MSD, and Bristol-Myers Squibb. JOW receives funding from CDC via contracts to his institution. LKM, FH and SJ have no conflicts to declare.] This does not alter our adherence to PLOS ONE policies on sharing data and materials.

years by 2030. A 12-month disruption to HCV treatment (20% reduction) and risk behaviors (25%,50%,75% reduction) during the COVID-19 pandemic would result in a 15% relative increase in total HCV incidence in 2030 compared to that expected under the status quo.

## Conclusions

HCV elimination among MSM living with HIV in Germany requires further DAA scale-up among those newly diagnosed combined with efforts to treat those previously diagnosed but untreated. Prospective monitoring will establish whether Germany is on track for HCV microelimination.

## Introduction

Hepatitis C virus (HCV) infection can lead to liver cirrhosis and death, but these conditions can be accelerated among individuals living with human immunodeficiency virus (HIV) infection [1]. Among men who have sex with men (MSM) living with HIV infection, the global burden of HCV is estimated to be 6.3% [2]. In Europe, HCV prevalence is 8.7 times higher among MSM living with HIV compared to HIV-negative MSM [2]. Similar HCV burden has been observed among MSM living with HIV in Germany, with an estimated HCV prevalence of 8.8% [2]. MSM living with HIV in Germany are also estimated to be nearly 21 times more likely to be coinfected with HCV compared to HIV-negative MSM and 15 times more likely compared to the general population [2]. Between 1990 and 2014, increasing trends in HCV incidence were observed across Europe [3]. Despite high HCV treatment rates with interferon-based regimens, HCV incidence among MSM living with HIV in Germany increased prior to HCV direct-acting antivirals (DAAs; [4]). Although MSM living with HIV and HCV were often initiated onto treatment with interferon-based regimens in the early phase of infection, cure rates with interferon-based regimens were insufficient and adverse events often limited treatment uptake [5]. Moreover, ongoing sexual risk-behavior and concomitant drug use ("Chemsex") contributed to ongoing HCV transmission in MSM.

In 2015, the World Health Organization (WHO) strategized global viral hepatitis elimination targets [6]. For HCV, these targets include an 80% reduction in HCV incidence and a 65% reduction in HCV-related mortality between 2015 and 2030 [6]. As current HCV DAAs can achieve >90% cure rates among individuals with HIV/HCV coinfection [7] with low side effect related discontinuation rates, treatment scale-up with DAAs offers a promising strategy to achieve the WHO elimination goals. However, Germany's progress towards these goals needs to be monitored to ensure current approaches will achieve these elimination goals, as the COVID-19 pandemic has resulted in widespread disruptions and the extent of their impact on HCV testing and treatment rates in Germany are still being examined. Especially, the impact of the pandemic on high-risk populations is unknown.

In 2019, a multicentric cohort on recently acquired HCV in MSM in Germany (NoCo) was established, using retrospective data since 2014, with plans to follow prospectively until 2022 to monitor the progress towards HCV microelimination among MSM. Using baseline retrospective data from this cohort, we developed a modeling framework to evaluate progress towards HCV elimination among MSM living with HIV in Germany and explored whether existing DAA treatment rates are sufficient to achieve the WHO HCV elimination incidence target by 2030. Further, we examined potential trajectories of progress towards HCV

elimination due to disruptions in HCV treatment [8] and reductions in risk behavior between 2020 and 2021.

## Methods

We adapted a previously published dynamic, deterministic model of HCV transmission, progression, and treatment among MSM living with diagnosed HIV to evaluate progress towards HCV elimination in Germany (**S1 Fig**; [9]). As we adapted a dynamic model, the risk of acquiring HCV was related to HCV prevalence and risk behavior. This model also included dynamic HCV transmission among MSM living with diagnosed HIV, fixed HCV incidence including infections acquired from HIV-undiagnosed or HIV-uninfected MSM outside of the MSM living with diagnosed HIV population. Individuals entered the model at the time of HIV diagnosis. A proportion had an existing HIV/HCV coinfection upon entry into the model. The model was stratified by HCV diagnosis status, HCV disease stage and treatment history, and high/low transmission risk. The high-risk group, characterized by the size of high-risk group and relative risk, was calibrated to epidemiological data for HCV primary infection and reinfection rates (**Table 1**). We assumed that the proportion of high-risk, MSM living with HIV remained stable but allowed for proportional mixing between groups and transitioning between risk groups. MSM living with HIV whose previous interferon-based therapies failed or became reinfected were eligible for retreatment. Given that there are no retreatment restrictions in Germany, MSM living with HIV in whom DAA treatment failed were also eligible to be retreated, at a rate equivalent to the primary treatment rates. All MSM living with HIV had a risk of mortality due to HIV and unrelated causes. An additional HCV-related mortality risk was included for MSM living with HIV coinfected with HCV.

### Model parameterization and calibration

The model was calibrated to and parameterized by historical epidemiological data on the HCV epidemic among MSM living with HIV in Germany (**Table 1**), using data from the HIV Seroconverter Cohort (a nationwide, multicenter prospective cohort study of 1,843 MSM living with diagnosed HIV in Germany between 1996 and 2012; [4]) and a German national cohort (NoCo) of patients from six HIV and hepatitis treatment sites (>8,000 MSM living with HIV between 2014 and 2020; [10]). In Germany, HCV incidence among MSM living with HIV increased from 1996–2012 (from ~0.5/100py to 2.8/100py; [4]), 8.2% seroprevalence among MSM living with HIV in 2012 [4], and stable HCV reinfection rates among MSM living with HIV in the pre-DAA and DAA era (6.82/100py from 2002–2014 and 7.33/100py from 2014–2018 [11]. Data from the NoCo study indicated that among MSM with a recently acquired HCV infection from 2014–2020, DAA treatment was initiated a median of 6 months after diagnosis in 81% (n = 148/182) of MSM who did not spontaneously clear their infection [10]. Additionally, 100% of MSM living with diagnosed HIV treated with DAAs achieved sustained viral response (SVR) [12].

The model was calibrated using an approximate Bayesian computation with sequential Monte Carlo scheme (ABC SMC) [13] for a resulting sample of 1,000 parameter sets (prior and posterior parameter ranges shown in **S1 Table**). To apply the ABC SMC methods, we sampled a set of parameters from a prior distribution, which was then used to generate a new dataset that is compared to observed data through a distance function. The final parameter set was a sample from the distribution, which given a small tolerance was a good approximation of the posterior distribution. A parameter set was considered acceptable when the distance or the log-likelihood, between the generated and observed data was less than the predetermined tolerance and resulted in the best approximation possible or was within the expected range for

**Table 1. Model parameterization and sources.**

| Parameters | Value | Reference |
|---|---|---|
| **Year of HCV epidemic seeding** | 1996 | [34] |
| **HCV testing rate per year** | Twice yearly from 2003 | [5] |
| **Duration from diagnosis to treatment (if treated)** | 6 months | |
| **SVR with DAAs** | 100% | [11, 35] |
| **Calibration parameters** | **Value (95% CI)** | **Reference** |
| **HCV primary incidence among diagnosed MSM living with HIV (by year)** | **1996–99**: 0.33 (95% CI: 0.05–2.34) | [4] |
| | **2000–03**: 0.47 (95% CI: 0.15–1.46) | |
| | **2004–07**: 0.94 (95% CI: 0.61–1.46) | |
| | **2008–12**: 2.28 (95% CI:1.79–2.89) | |
| **HCV prevalence (Ab+ or RNA+) among diagnosed MSM living with HIV in 2012** | 8.2% (95% CI: 7.0–9.5) | [4] |
| **HCV reinfection incidence after treatment or spontaneous clearance 2002–2014** | **2002–14**: 8.2/100 py (95% CI 5.6–12.1) | [15, 36] |
| **Number of MSM living with HIV in Germany in 2015** | 53,800 (95%CI 49,800–58,500) | [37] |
| **Parameters varied for fitting** | **Value (sampling range, distribution)** | **Comments** |
| **Proportion of HIV-positive who spontaneously clear acute HCV infection** | 11% (9–15%, uniform) | [5, 38, 39] |
| **Proportion of those infected who do not spontaneously clear initiated onto treatment within 6 months of diagnosis** | 80% (excluding those who spontaneously clear the virus) from 2002 (75–85%, uniform) | [16] |
| **Proportion of MSM living with HIV infected with HCV upon HIV diagnosis** | 0.65% (0.35–0.95%, uniform) | [4] |
| **Duration acute infection until spontaneous clearance** | 6 months (3–9 months, uniform) | [39] |
| **SVR with IFN/RBV** | | |
| **<1 year from HCV infection** | 70% (65–75%, uniform) | [40] |
| **>1 year from HCV infection** | 30% (25–35%, uniform) | Weighted based on genotype distribution and SVR by genotype from a recent meta-analysis [41] |
| **Life expectancy from HIV diagnosis** | Varies over calendar time based on increasing ART coverage and earlier diagnosis (20–40 years from assumed HIV diagnosis and ART initiation at age 35) | [17, 42–47] |
| **Excess liver-related mortality due for those with chronic HCV (annual)** | 0.16 per 100 person-years (0.05–0.27, uniform) | [18, 19] |
| **Background HCV incidence from outside MSM living with diagnosed HIV population** | 1.5/1,000 person-years (0-2/1,000 person-years, uniform) | Assumed similar to observed in HIV-negative MSM population [20] |
| **Proportion high risk** | 0–30%, uniform | Among a sub-sample of a large Internet survey among MSM in Europe in 2010, 5% of all MSM in Berlin reported consumption of drugs typically used at sex parties (ecstasy, amphetamines, crystal methamphetamine, mephedrone, GHB/GBL, ketamine, or cocaine) in the preceding 4 weeks, but MSM living with HIV were 5-times more likely to report this risk [48]. A German study among MSM living with HIV in 2014, reporting that 17% of MSM living with HIV report recent substance use [49]. |
| **Relative risk high risk compared to low risk** | 0–100, uniform | Fitted mean value higher than relative risks of HCV infection among MSM living with HIV in Germany with associated individual behaviors [50–51]. |
| **Leaving rate from high risk (annual)** | 0–0.5, uniform | |

*(Continued)*

**Table 1.** (*Continued*)

| Initial HCV prevalence in 1996 | | |
|---|---|---|
| Low risk | 0–1%, uniform | |
| High risk | 0–1%, uniform | |
| **Infection rate** | 0–0.2, uniform | |
| **Number of MSM living with diagnosed HIV in 1996** | 1,000–15,000, uniform | |
| **Number of new entrants to MSM living with diagnosed HIV population each year** | 3,000–4,000, uniform | Includes new HIV-diagnoses and those previously diagnosed and migrating to Berlin |

HCV: hepatitis C virus, HIV: human immunodeficiency virus, MSM: men who have sex with men, Ab: antibody, RNA: ribonucleic acid, IFN/RBV: interferon and ribavirin, DAA: direct-acting antivirals.

total number of MSM living with HIV in Germany in 2015 (48,000–58,000) [14], HCV primary incidence (1996–1999, 2000–2003, 2004–2007, and 2008–2012), HCV seroprevalence among MSM living with diagnosed HIV in 2012 (7.0–9.5%) [4] and HCV reinfection incidence among MSM living with diagnosed HIV (5.6–12.1/100 person-years across 2002–2014) [15].

In agreement with German guidelines, the model incorporated biannual testing with an 80% initiation onto treatment within 6 months following HCV diagnosis from 2002 [5]. As HCV treatment shifted from interferon-based to interferon-free DAA therapy following German regulatory approval in mid-2014, we modeled this shift in treatments beginning in 2015. As high treatment rates (55–83%) have been observed across Germany between 2007–2015 ([16]), we modeled high HCV testing and treatment rates among MSM living with HIV. Annual all-cause mortality among MSM living with diagnosed HIV varied over calendar time based on increasing antiretroviral therapy (ART) coverage and earlier diagnosis (an estimated life expectancy increase of 20–40 years from assumed HIV diagnosis and ART initiation at age 35 years based on estimates from the UK, where age at HIV diagnosis and antiretroviral therapy coverage are similar to estimates in Germany) [17]), and also included annual, excess HCV-related liver-related mortality rates [18, 19]. As the contribution to HCV incidence from HIV-undiagnosed or HIV-uninfected MSM among MSM living with HIV in Germany remains unknown, we assumed a similar contribution to that observed among HIV-uninfected MSM (0-2/1,000 person-years[py]; [20]). Posterior estimates for background HCV incidence from outside the MSM living with diagnosed HIV population was 0.6/1,000py (95% CI 0.2–1.0/1,000py). All simulations were performed using Matlab R2020b software.

### Intervention scenarios

We modeled several scenarios including: (1) status quo, defined as no change in HCV treatment rates (80% of newly diagnosed HCV infections treated within 6 months after diagnosis); (2) status quo, with all newly diagnosed HCV infections treated within 3 months; (3) treatment scale-up to 90% in 2021 for all newly diagnosed HCV infections within 6 months; (4) treatment scale-up to 100% in 2021 for all newly diagnosed HCV infections within 6 months; (5) treatment scale-up to 100% beginning in 2021 for all newly diagnosed HCV infections within 6 months along with 15% per year of previously diagnosed and untreated infections; (6) as in scenario 4, but with all newly diagnosed HCV infections treated within 3 months.

### Potential impacts of the COVID-19 pandemic

While data collection on HCV incidence, testing, and treatment in the NoCo cohort is ongoing, we explored the potential impact of the COVID-19 pandemic on service disruptions by modeling a 20% disruption to treatment in 2020 [8] for 12 months beginning in 2020. In addition to the 20% disruption to treatment, we modeled reductions in risk behavior by 25%, 50%, and 75% for 12 months between 2020 and 2021 [21–23].

## Results

### Model fit to data

The model fit well to HCV primary incidence, defined as the incidence of first HCV infection, over time (**Fig 1A**). However, the model slightly overestimated the HCV seroprevalence (antibody-positive or RNA-positive) among MSM living with diagnosed HIV in 2012 (**Fig 1B**).

### 2021 HCV epidemic among MSM living with diagnosed HIV in Germany

Based on model estimates, there are approximately 31,969 (95% CI 28,791–35,537) MSM living with diagnosed HIV in Germany in 2021. Among MSM living with diagnosed HIV in Germany in 2021, overall HCV incidence is 3.1 per 100 person-years (95% CI 2.7–3.7/100py; **S2 Fig**) and primary HCV incidence is 2.3 per 100 person-years (95% CI 2.0–2.8/100py) (**Fig 2**). The model estimated HCV seroprevalence (antibody or RNA-positive) to be 21.1% (95% CI 19.1–23.4; **S3 Fig**) in 2021 and HCV chronic prevalence (RNA-positive) to be 7.8 (95% CI 6.8–9.0; **S4 Fig**), with a corresponding 5,170 (95% CI 4,475–5,960) MSM living with diagnosed HIV chronically infected with HCV.

### Impact of existing treatment

Existing annual treatment levels of ~80% within six months of diagnosis of MSM living with diagnosed HIV cannot achieve the WHO elimination target by 2030 (**Fig 3**). The current treatment strategy results in 7% relative reduction in total incidence by 2030 (95% CI -2.3–17.4). However, the current treatment strategy would have a stronger impact on primary HCV incidence resulting in a 26.6 (95% CI 17.7–34.7) incidence reduction between 2015–2030. Under this status quo, there would be no improvement in HCV seroprevalence (Ab or RNA-positive) or chronic HCV prevalence (RNA-positive) over time (**S5 Fig**). If treatment levels remained at

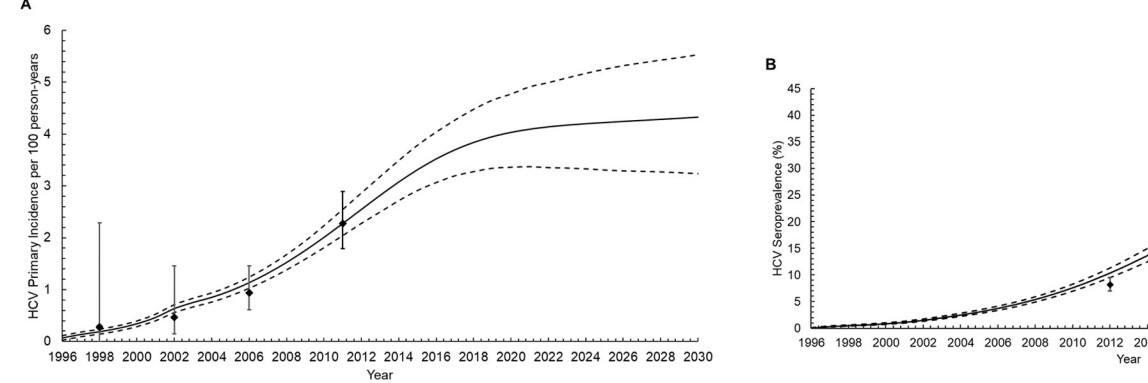

**Fig 1.** Model fit to calibration data for primary hepatitis C virus (HCV) incidence (A) and HCV seroprevalence (antibody or RNA positive) (B) among men who have sex with men living with diagnosed human immunodeficiency virus (HIV) in Germany. Diamonds represent mean epidemiological data estimate; whiskers, 95% confidence intervals; solid lines, the mean model trajectories; dashed lines, the 2.5% and 97.5% confidence percentiles of trajectories.

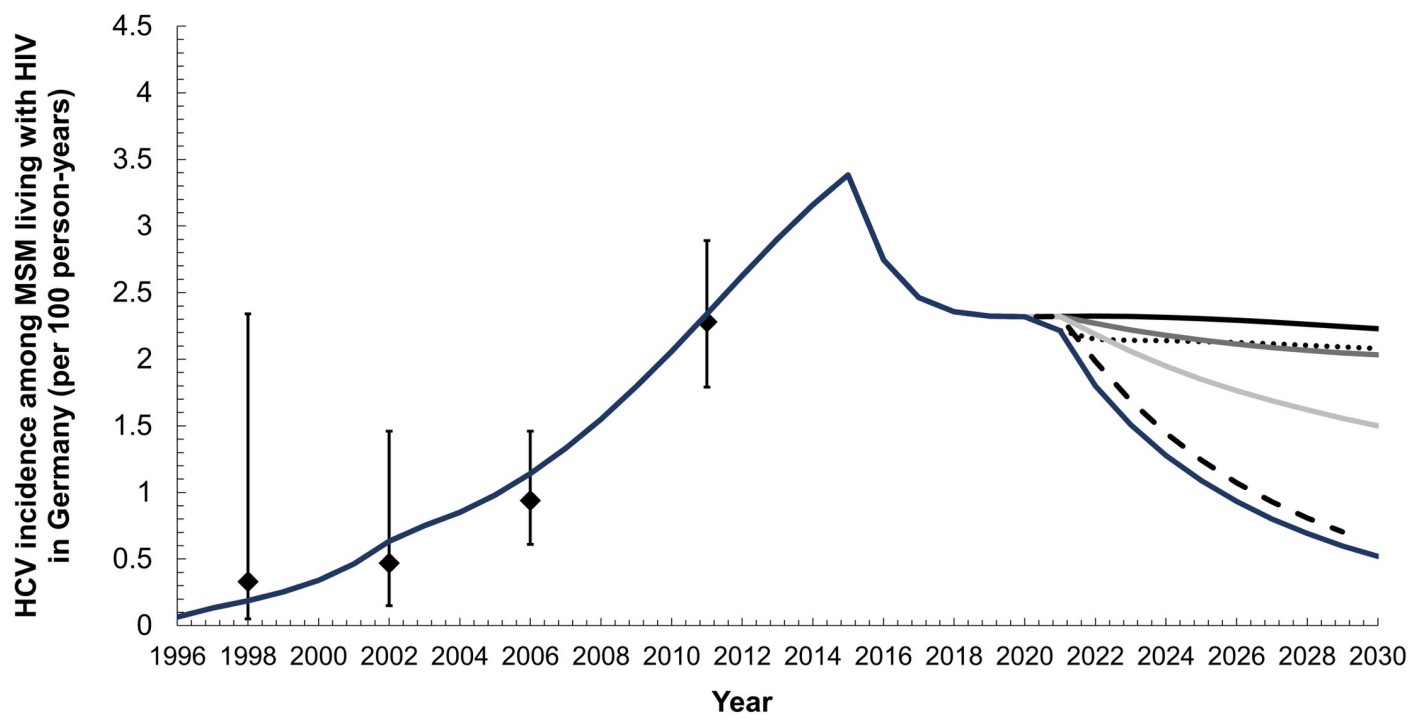

**Fig 2. Primary hepatitis C virus (HCV) incidence among men who have sex with men living with diagnosed human immunodeficiency virus (HIV) in Germany.** Diamonds represent mean epidemiological data estimate; whiskers, 95% confidence intervals; solid lines, the mean model trajectories. 'Early' refers to all newly diagnosed HCV infections treated within 3 months.

status quo, but the time between diagnosis and treatment was halved (3 months vs 6 months), at best, an 16.8% total incidence reduction (95% CI 7.0–28.1) could be achieved by 2030. Further, primary incidence under this earlier diagnosis strategy would have a similar result as status quo, resulting in 33.2% reduction (95% CI 25.8–40.1). While HCV seroprevalence would not improve over time (mean increase: 69.5%, 95% CI 57.9–80.8), reducing the time between diagnosis and treatment to 3 months could lead to a 11.5% reduction (95% CI 1.7–23.0) in HCV chronic incidence between 2015–2030.

## Impact of treatment scale-up

Under the current time to treatment from diagnosis (6 months), scaling-up treatment to 90% or 100% per year is not sufficient to achieve the WHO incidence target of 80% reduction between 2015 and 2030 (90% treatment scale-up: 21.8% [95% CI 10.6–32.3]; 100% treatment scale-up: 47.1% [95% CI 35.9–56.6]). Treating all (100%) at diagnosis per year would result in 56.6% (95% CI 52.9–60.5) reduction in primary HCV incidence (vs 26.3% [95% CI 17.7–34.7%] in status quo treatment level and 37.4% [95% CI 33.0–41.1%] in 90% treatment scale-

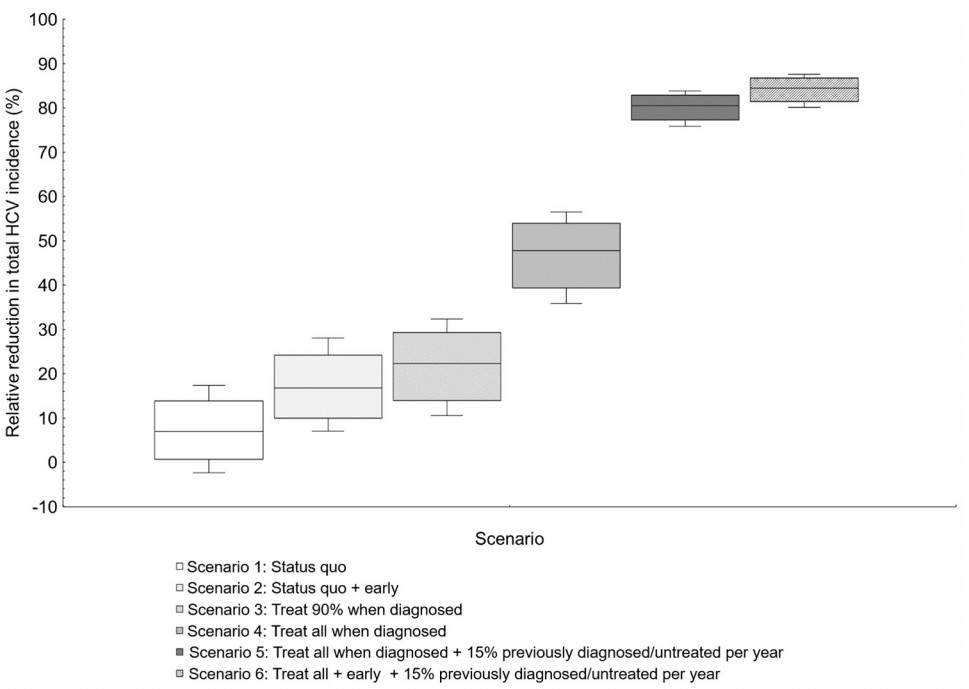

**Fig 3. Relative reduction in total hepatitis C virus (HCV) incidence between 2015–2030 among men who have sex with men living with diagnosed human immunodeficiency virus (HIV) in Germany.**

up). Under the treat all strategy would result in a 98% greater decrease in HCV chronic prevalence compared to the status quo treatment level (44.3% vs 0.7%; **S5 Fig**).

## Strategies to achieve WHO HCV elimination incidence target

Modeling indicated that under the existing treatment program (~80% of MSM living with HIV per year), the WHO HCV elimination incidence target cannot be met (**Fig 3**). Further, a treatment level of 100% in combination with treating at least 15% of previously diagnoses and untreated MSM living with HIV (Scenario 5 and 6) is required to achieve an incidence reduction of 80% between 2015 and 2030. Scenario 5 and 6 resulted in the most dramatic decrease in incidence between 2015 and 2030 (Scenario 5: 80.2% [95% CI 75.8–83.8]; Scenario 6: 84.3% [95% CI 80.1–87.7]). These strategies would also result in substantial reductions in HCV primary incidence (Scenario 5: 82.6% [95% CI 80.6–84.4]; Scenario 6: 85.8% [95% CI 83.6–87.9]) and chronic prevalence (Scenario 5: 81.1% [95% CI 78.8–83.4]; Scenario 6: 84.5% [95% CI 82.5–86.4]).

## Potential impact of the COVID-19 pandemic on HCV incidence target

Potential impacts of the HCV testing and treatment service disruptions on HCV total incidence is shown in **Fig 4.** All modeled disruptions resulted in increased HCV incidence between 2021 and 2030. Assuming treatment rates decreased by 20% beginning in 2020 but rebounded within 12 months to status quo levels, after a 12% relative increase in HCV incidence in 2020 to 2021, total HCV incidence would be 15% greater than expected under the status quo treatment regimen by 2030 (4.7/100py vs 4.0/100py). However, treatment disruptions at 20% for 12 months between 2020 and 2021 in addition to reductions in risk behaviors by 75%, would temporarily reduce incidence to 0.8/100py in 2021, but would quickly rebound

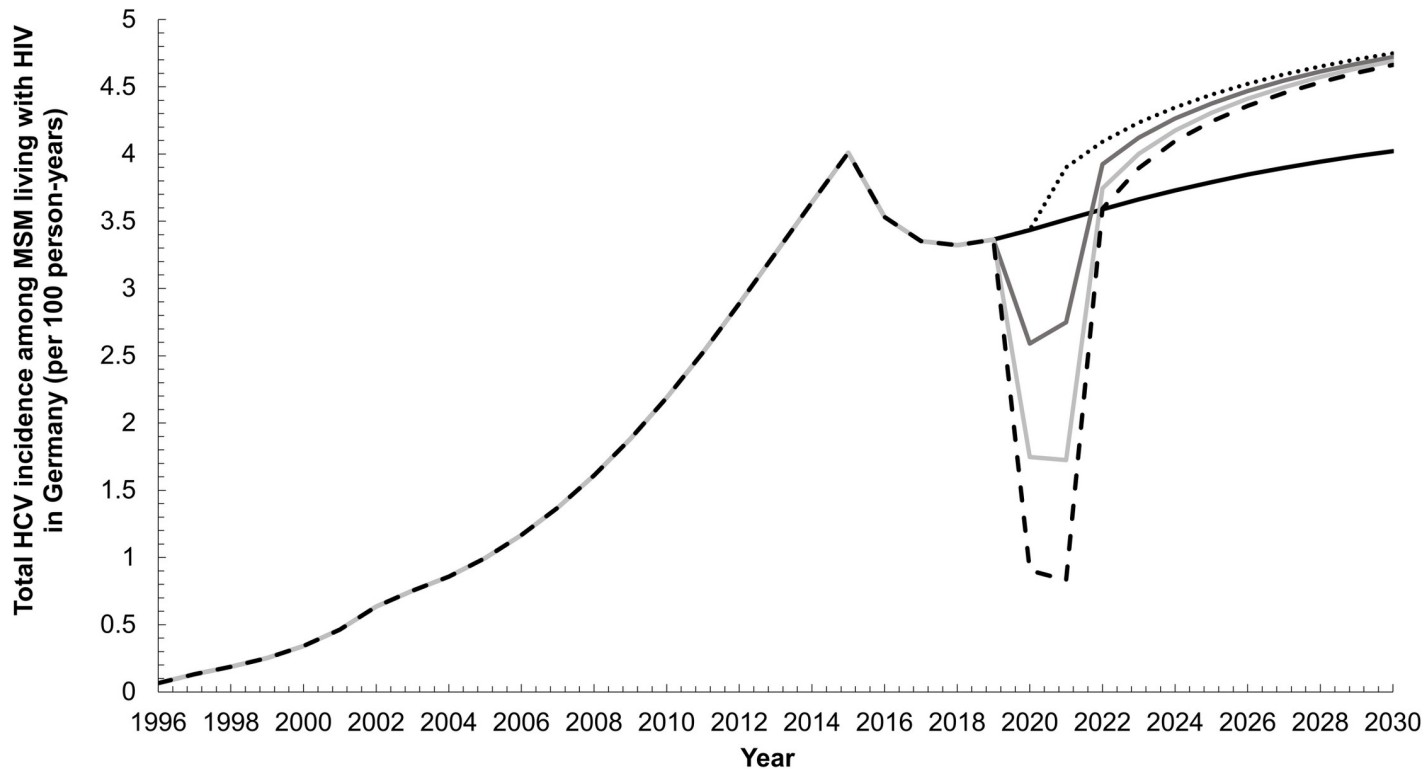

**Scenario 1: Status quo**
**Scenario 2: 20% reduction in treatment for 12 months**
**Scenario 3: 20% reduction in treatment + 25% reduction in risk behavior for 12 months**
**Scenario 4: 20% reduction in treatment + 50% reduction in risk behavior for 12 months**
**Scenario 5: 20% reduction in treatment + 75% reduction in risk behavior for 12 months**

**Fig 4. Modeled disruptions in hepatitis C virus (HCV) screening, treatment, and various reductions in risk behavior to epidemic trajectories of total HCV incidence among men who have sex with men living with diagnosed HIV in Germany.**

surpassing expected HCV incidence under the status quo by 2023 (3.9/100py vs 3.7/100py). Less severe disruptions to risk behavior (25% and 50%) show similar results, with rapid reductions in incidence in 2020 but surpassing status quo incidence rates by 2022 and continuing to climb through 2030. Without further intervention or treatment scale-up, the HCV incidence goal would not be met.

## Discussion

This modeling study examines the current state of the HCV epidemic among MSM living with HIV in Germany and provides a baseline assessment to explore what treatment and combination scale-up strategies could achieve HCV microelimination among people with HIV by 2030. Our model projections show that further treatment scale-up is required to achieve the WHO incidence target among MSM living with HIV in Germany by 2030. Furthermore, treatment scaled-up to 100% at diagnosis and in the absence of treating previously HCV-diagnosed and untreated MSM living with HIV could not achieve the incidence reduction by 2030. However, reaching the HCV incidence elimination target is possible through the implementation of combination strategies which shorten the time between diagnosis and treatment and treat previously diagnosed and untreated. We note that retention in care strategies remain

important in the HCV elimination era. Retention to HCV care appears high among MSM living with diagnosed HIV/HCV (with >80% of MSM living with HIV initiated onto HCV treatment). Further, timely diagnosis of HCV among MSM living with diagnosed HIV is facilitated by high retention in HIV care (>93% of PLHIV are on ART [24]). Nevertheless, a high proportion of individuals living with diagnosed HIV experience gaps in care (roughly 2/3 have experienced a gap, with a median time per gap of 223 days in Germany [25]). If MSM are infected during one of these gap periods, this could delay HCV diagnosis and treatment and be an obstacle to HCV elimination. This underscores the importance of exploring strategies which do not solely rely on treatment alone but may require programmatic changes which has also been explored in other settings in Europe such as the United Kingdom [26]. Our findings support early treatment for HCV under the current guidelines and that reimbursement of early treatment will be needed to support this approach. Further, disruptions in HCV screening and treatment during 2020 could delay progress towards these goals and must be examined to understand the extent to which these disruptions impacted HCV elimination efforts. The prospective monitoring of the NoCo cohort will provide subsequent data with which this model can be used to assess whether Germany is on track to reach the HCV incidence and mortality targets among people with HIV by 2030.

## Comparisons with existing literature

Our study supports results from other HCV elimination modeling studies in the United Kingdom which showed that earlier access to DAA treatment and treatment for reinfection among MSM living with diagnosed HIV could lead to substantial reductions in both primary and total HCV incidence [26] and in the Netherlands which showed that universal access to DAA treatment and availability increased uptake and subsequently decreased HCV incidence among MSM living with HIV [27]. Further, in Switzerland where similar increases in HCV infection were observed among MSM living with HIV between 2002–2011 [28], reductions in both primary and total HCV incidence were reported from the Swiss HCVree trial which performed routine clinical evaluation every 3 to 6 months, annual HCV antibody screening, HCV PCR screening, DAA treatment and behavioral interventions and rescreening following intervention completion [29]. Our modeling results build on previous analyses among MSM living with diagnosed HIV in Berlin, which found that a combination DAA scale-up with moderate behavioral interventions were required to achieve the HCV elimination incidence target by 2030 [30]. While our results support findings from other modeling studies in high-income settings with increasing HCV infections, our findings contrast with a modeling study in Australia among MSM living with HIV found that modest scale-up of DAAs is sufficient to achieve elimination [31]. However, this study assumed that risk behaviors would become stable and thus may not be extended to MSM living with HIV in other settings.

## Limitations

Our study has several limitations. First, as with all modeling studies there is uncertainty in the parameterization and model projection. To account for this, we utilized approximate Bayesian calibration methods, which is a robust method to incorporate uncertainty within our model as well as variations in our model simulations.

Second, the impact of the COVID-19 pandemic on HIV and HCV services, HCV elimination efforts, risk groups, and transmission is not fully understood at this time. While preliminary data from NoCo indicates that MSM living with diagnosed HIV attending HIV clinics in Germany, experienced near normal service delivery and care during the pandemic period, this only provides a snapshot into the services provided to MSM living with diagnosed HIV who

access, attend, and receive care from these HIV and HCV clinics. It is to date unknown, to what extent the pandemic and/or its containment measures have impacted HCV testing or treatment rates. As HCV in MSM occurs in a setting of sexual risk behavior and drug use, it is further unknown, in which way these behavioral patterns may have been shifted during the pandemic.

Third, our assumptions on the proportion of high and low risk were based on pre-pandemic estimates. It is unclear whether the pandemic has increased or decreased risk behaviors among MSM living with diagnosed HIV in Germany during this period though ongoing observational and clinical data collection will add to our understanding of these impacts over time. Among HIV-negative MSM in the UK, high rates of sexual activity, sexually transmitted infections, and increased challenges to accessing sexual health services were observed during the lockdown [32]. While reduced uptake of pre-exposure prophylaxis (PrEP) was observed among MSM in Australia, this was accompanied by decreased rates of sexual activity contrary to observations in the UK [33]. Thus, it is unclear if similar trends could be assumed of MSM living with HIV and therefore were not reflected in our assumptions. However, using NoCo data collected during the COVID-19 pandemic era, we will probe these areas of uncertainty further using our modeling framework developed here.

Finally, our study explored treating 15% of previously diagnosed and untreated infections per year, based on expected feasibility. Though this was informed by data from the NoCo cohort of MSM living with HIV and HCV in Germany, further research could explore feasibility of treating more than 15% of previously diagnosed and untreated infections per year.

## Applying the framework to understand COVID-19 screening and treatment disruptions

As this framework has been informed by the NoCo study, HCV incidence data collected during the COVID-19 pandemic is forthcoming and will be used to validate our COVID findings. Using this framework, we will further evaluate the impact of the COVID-19 pandemic on HCV elimination progress among MSM living with HIV in Germany by collecting prospective clinical, biological, and behavioral data to explain the success of the intervention uptake within the NoCo cohort.

## Conclusions

Baseline HCV elimination progress shows that treatment and treatment scale-up alone is not sufficient to achieve the WHO incidence goal among HIV-diagnosed in Germany between 2015 and 2030. Other prevention strategies which decrease the time between diagnosis and treatment and explore treating individuals who were previously diagnosed but untreated are necessary. Finally, it is necessary to understand the impact and disruptions arising from the COVID-19 pandemic on testing and treatment uptake to appropriately evaluate the current progress towards HCV elimination among MSM living with HIV in Germany.

## Supporting information

**S1 Table. Prior and posterior ranges for parameters which were varied for model fitting.** HCV: hepatitis C virus; MSM: men who have sex with men; SVR: sustained virological response; IFN/RBV: Interferon/Ribavirin; UK: United Kingdom.
(DOCX)

**S1 Fig. Adapted HCV transmission model schematic among MSM living with diagnosed HIV.**
(TIF)

**S2 Fig. Modeled epidemic trajectories of total HCV incidence (primary and reinfection) among MSM living with diagnosed HIV in Germany.** Primary incidence is defined as the incidence of first HCV infection. 'Early' refers to all newly diagnosed HCV infections treated within 3 months.
(TIF)

**S3 Fig. Modeled epidemic trajectories of HCV seroprevalence (antibody or RNA positive) among MSM living with diagnosed HIV in Germany.** Data point is HCV seroprevalence estimate of 8.2% in 2012. 'Early' refers to all newly diagnosed HCV infections treated within 3 months.
(TIF)

**S4 Fig. Modeled epidemic trajectories of HCV chronic prevalence (RNA positive) among MSM living with diagnosed HIV in Germany.** 'Early' refers to all newly diagnosed HCV infections treated within 3 months.
(TIF)

**S5 Fig. Relative reductions in HCV seroprevalence and total HCV incidence among MSM living with diagnosed HIV in Germany.** A). Relative reductions in total HCV incidence among MSM living with diagnosed HIV in Germany between 2015–2030. B). Relative reductions in primary HCV incidence among MSM living with diagnosed HIV in Germany between 2015–2030. C). Relative reductions in HCV chronic prevalence among MSM living with diagnosed HIV in Germany between 2015–2030. HCV chronic prevalence is defined as RNA + only. HCV: hepatitis C; MSM: Men who have sex with men. Primary incidence is defined as the incidence of first HCV infection. 'Early' refers to all newly diagnosed HCV infections treated within 3 months.
(TIF)

## Author Contributions

**Conceptualization:** Patrick Ingiliz, Natasha K. Martin.

**Data curation:** Patrick Ingiliz, Sonia Jain, Feng He.

**Formal analysis:** Lara K. Marquez.

**Funding acquisition:** Patrick Ingiliz.

**Investigation:** Patrick Ingiliz.

**Methodology:** Lara K. Marquez, Natasha K. Martin.

**Supervision:** Patrick Ingiliz, Jürgen K. Rockstroh, Joel O. Wertheim, Natasha K. Martin.

**Validation:** Lara K. Marquez.

**Visualization:** Lara K. Marquez.

**Writing – original draft:** Lara K. Marquez, Patrick Ingiliz, Natasha K. Martin.

**Writing – review & editing:** Lara K. Marquez, Patrick Ingiliz, Christoph Boesecke, Ivanka Krznaric, Knud Schewe, Thomas Lutz, Stefan Mauss, Stefan Christensen, Jürgen K. Rockstroh, Sonia Jain, Feng He, Joel O. Wertheim, Natasha K. Martin.

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
