## [Decision Letter · Decision Letter 0]

14 Mar 2022

PONE-D-22-03984Establishing a framework towards monitoring HCV microelimination among HIV-positive men who have sex with men in Germany: A modeling analysisPLOS ONE

Dear Dr. Marquez,

Thank you for submitting your manuscript to PLOS ONE. After careful consideration, we feel that it has merit but does not fully meet PLOS ONE’s publication criteria as it currently stands. Therefore, we invite you to submit a revised version of the manuscript that addresses the points raised during the review process.

We look forward to receiving your revised manuscript.

Kind regards,

Chen-Hua Liu

Academic Editor

PLOS ONE

Journal Requirements:

[NKM has received unrestricted research grants and honoraria from Gilead and Merck. CB reports personal fees from AbbVie, Gilead, Johnson & Johnson, Merck Sharp & Dohme (MSD), and ViiV and grants from DZIF (German Center for Infection Research), Deutsche Leberstiftung, and Hector Stiftung. TL reports grants from AbbVie, Gilead Sciences, ViiV, Janssen-Cilag, MSD, Deutsche Leberstiftung, and DAGNÄ for scientific work. SC reports personal fees from Gilead, AbbVie, ViiV, MSD, and Indivior. S. M. reports personal fees from AbbVie, Gilead, and MSD. JKR reports personal fees from Abivax, AbbVie, Gilead, Janssen, Merck, and ViiV, outside the submitted work. PI has received unrestricted research grants from Gilead (IN-DE-987-4637 and IN-FR-980-6332), a research grant from Abott Laboratories, and speaker fees from Gilead, AbbVie, ViiV, MSD, and Bristol-Myers Squibb. JOW receives funding from CDC via contracts to his institution. LKM, FH and SJ have no conflicts to declare.] 

Reviewers' comments:

Reviewer's Responses to Questions

**Comments to the Author**

1. Is the manuscript technically sound, and do the data support the conclusions?

Reviewer #1: Yes

Reviewer #2: Yes

2. Has the statistical analysis been performed appropriately and rigorously? 

Reviewer #1: Yes

Reviewer #2: Yes

3. Have the authors made all data underlying the findings in their manuscript fully available?

Reviewer #1: Yes

Reviewer #2: Yes

4. Is the manuscript presented in an intelligible fashion and written in standard English?

Reviewer #1: Yes

Reviewer #2: Yes

5. Review Comments to the Author

Reviewer #1: This manuscript described “Establishing a framework towards monitoring HCV microelimination among HIV-positive men who have sex with men in Germany: A modeling analysis.” The result presents that HCV elimination among HIV+ MSM in Germany requires further DAA scale-up among those newly diagnosed combined with efforts to treat those previously diagnosed but untreated.

In general, this is a well-written manuscript. Other points in this manuscript needed to be clarified are listed below:

1. HIV+MSM : I will suggest “MSM living with HIV” in aspect of terminology

Minor revisions:

1. Page 4, line 83: the subject is missing, Although “what” often initiated…

2. Page 13, line 202: please make sure the number of confidence interval is correct 2.5% and 97.5% ?

3. References: please unify the standard form of references, such number of authors, doi, PMID, etc….

Reviewer #2: 1. The modeling presumed that all HIV+MSM are well monitored with regular HCV testing. Considering the retention in care, will loss to follow up in HIV care might affect HCV elimination?

2. Is there any reason that scenario 5 choose to treat “15% per year” of previously diagnosed and untreated infections!

3.Recently, some experts use the pool serum to detect the relapse rate of HCV among the groups including previously DAA treatment experiences HCV patients with elevated liver function test, sexual transmitted disease and found some relapses cases! Such results will influence the HCV elimination in future! Does our model can predict or avoid such influences?

4. For HIV infected patients, same day therapy seems to be helpful to reduce the incidence! According to your model, if we further shortening the duration of diagnosis to treatment of HCV, will it be more helpful to achieve the goal of elimination of HCV infection?

6. PLOS authors have the option to publish the peer review history of their article (what does this mean?). If published, this will include your full peer review and any attached files.

Reviewer #1: **Yes: **CHENG CHIEN-YU

Reviewer #2: No

---

## [Author Response · Author response to Decision Letter 0]

13 Apr 2022

Dear Editor, 

We would like to submit the following publication “Establishing a framework towards monitoring HCV microelimination among men who have sex with men living with HIV in Germany: A modeling analysis,” (PONE-D-22-03984) for re-consideration as an original research article in PLOS ONE. We have revised the article based on the comments from reviewers. We thank the reviewers for their comments and have responded to the comments point by point below, with the changes tracked in the submitted version. We believe the manuscript has been strengthened with the suggested edits. Additional journal requirements noted in the revision letter for competing interests and financial disclosures have been included in the cover letter. Please let us know if there are any further questions or concerns.

Sincerely,

Lara Marquez, on behalf of the coauthors

This manuscript described “Establishing a framework towards monitoring HCV microelimination among HIV-positive men who have sex with men in Germany: A modeling analysis.” The result presents that HCV elimination among HIV+ MSM in Germany requires further DAA scale-up among those newly diagnosed combined with efforts to treat those previously diagnosed but untreated. 

In general, this is a well-written manuscript. Other points in this manuscript needed to be clarified are listed below:

Reviewer #1, Comment #1: HIV+MSM: I will suggest “MSM living with HIV” in aspect of terminology 

Author reply: We thank the reviewer for their comments and agree that this terminology is much improved by this suggestion. We have incorporated this throughout the manuscript, figures, and supplementary information, replacing any occurrence of ‘HIV+ MSM’ with ‘MSM living with HIV.’

Reviewer #1, Minor revisions comment #1: Page 4, line 83: the subject is missing, Although “what” often initiated…

Author reply: We have revised this sentence to be more clear. 

Introduction, page 4, lines 86-88: “Although MSM living with HIV and HCV were often initiated onto treatment with interferon-based regimens in the early phase of infection, cure rates with interferon-based regimens were insufficient and adverse events often limited treatment uptake [5].”

Reviewer #1, Minor revisions comment #2: Page 13, line 202: please make sure the number of confidence interval is correct 2.5% and 97.5%?

Author reply: We confirm that we have shown the 2.5%-97.5% confidence boundaries but have revised the text slightly to note confidence percentile for these bounds, rather than interval. 

Results, page 13, lines 210-212: “Diamonds represent mean epidemiological data estimate; whiskers, 95% confidence intervals; solid lines, the mean model trajectories; dashed lines, the 2.5% and 97.5% confidence percentiles of trajectories.”

Reviewer #1, Minor revisions comment #3: References: please unify the standard form of references, such number of authors, doi, PMID, etc…. 

Author reply: We have revised the references (On pages 20-28) to be uniform throughout. 

Reviewer #2, Comment #1: The modeling presumed that all HIV+MSM are well monitored with regular HCV testing. Considering the retention in care, will loss to follow up in HIV care might affect HCV elimination?

Author reply: We appreciate this question from the reviewer. Within this cohort in Germany, between 2014-2020, HCV treatment coverage with direct acting antiviral therapy was >80% among MSM living with HIV who did not spontaneously clear HCV infection and therefore believe dropout rate to be low. However, despite a high proportion of MSM living with HIV receiving antiretroviral therapy (ART; UNAIDS, 2021), recent evidence from two large, multicenter cohort studies among PLHIV in Germany showed that 93% of people living with diagnosed HIV were on ART (an der Heiden et al., 2019) and further nearly 2/3 of PLHIV initiated onto ART have experienced a gap (median time per gap in Germany: 223 days; Schmidt et al. 2021). Given this high proportion of PLHIV experiencing at least one gap in HIV care and in receiving ART, HCV diagnosis and treatment could be delayed if an individual is infected with HCV during this gap period. As such, this could delay HCV elimination efforts and have added new text in the discussion section highlighting this. 

Discussion, page 17, lines 300-308: “We note that retention in care strategies remain important in the HCV elimination era. Retention to HCV care appears high among MSM living with diagnosed HIV/HCV (with >80% of MSM living with HIV initiated onto HCV treatment). Further, timely diagnosis of HCV among MSM living with diagnosed HIV is facilitated by high retention in HIV care (>93% of PLHIV are on ART [an der Heiden et al., 2019]). Nevertheless, a high proportion of individuals living with diagnosed HIV experience gaps in care (roughly 2/3 have experienced a gap, with a median time per gap of 223 days in Germany [Schmidt et al. 2021]). If MSM are infected during one of these gap periods, this could delay HCV diagnosis and treatment and be an obstacle to HCV elimination.”

Reference: 

1. UNAIDS. Germany HIV and AIDS Estimates. 2021. Accessed on 04 April 2022 from https://www.unaids.org/en/regionscountries/countries/germany

2. an der Heiden M, Marcus U, Kollan C, Schmidt D, Gunsenheimer-Bartmeyer B, Bremer V. Schätzung der Zahl der HIV-Neuinfektionen und der Gesamtzahl von Menschen mit HIV in Deutschland, Stand Ende 2018. Epidemiologisches Bulletin. 2019;(46):483–92.

3. Schmidt, D., Kollan, C., Stoll, M. et al. Everything counts - a method to determine viral suppression among people living with HIV using longitudinal data for the HIV care continuum - results of two large, German, multi-center real-life cohort studies over 20 years (1999–2018). BMC Public Health 21, 200 (2021). https://doi.org/10.1186/s12889-020-10088-7

Reviewer #2, Comment #2: Is there any reason that scenario 5 choose to treat “15% per year” of previously diagnosed and untreated infections!

Author reply: We selected to treat 15% of previously diagnosed and untreated infections per year based on what was feasible and achievable in this setting. This was informed by data from the NoCo Cohort of MSM living with HIV and HCV in Germany. We have added further clarification in text for this. 

Discussion, Limitations, page 19, lines 364-367: “Finally, our study explored treating 15% of previously diagnosed and untreated infections per year, based on expected feasibility. Though this was informed by data from the NoCo cohort of MSM living with HIV and HCV in Germany, further research could explore feasibility of treating more than 15% of previously diagnosed and untreated infections per year.”

Reviewer #2, Comment #3: Recently, some experts use the pool serum to detect the relapse rate of HCV among the groups including previously DAA treatment experiences HCV patients with elevated liver function test, sexual transmitted disease and found some relapses cases! Such results will influence the HCV elimination in future! Does our model can predict or avoid such influences?

Author reply: Our model accounts for both acute infection and reinfections (including those previously treated with DAAs). As such, we incorporate the potential impact of reinfections on reducing the cost-effectiveness of testing and treatment. Due to the risk of reinfection, MSM with HIV are recommended for routine testing after SVR. We agree with the reviewer that pooled testing could be used for surveillance in this group, however we note that pooled testing is most useful in situations where the prevalence of infection is low and therefore it is more efficient and cost-effective to pool. Because incidence of reinfection can be high among MSM previously infected with HCV (Ingiliz et al., 2017), pooling could unnecessarily delay timely identification of infection in this group and may not be more efficient than unpooled testing. Analyses examining the value of pooled testing in relation to prevalence of infection in a subgroup and time to pool would be useful to assess the relative benefits of this strategy among MSM.

References: 

1. Ingiliz P, Martin TC, Rodger A, Stellbrink H-J, Mauss S, Boesecke C, et al. HCV reinfection incidence and spontaneous clearance rates in HIV-positive men who have sex with men in Western Europe. J Hepatol. 2017;66(2):282–7. Available from: https://doi.org/10.1016/j.jhep.2016.09.004

2. Aguilera A, Pereira S, Fuentes A, de Salazar A, Trastoy R, Navarro D, Picchio CA, Lazarus JV, García F. Pooling samples for hepatitis C RNA detection. Lancet Gastroenterol Hepatol. 2021 Aug;6(8):608-609. doi: 10.1016/S2468-1253(21)00217-X. PMID: 34246354; PMCID: PMC8266286.

Reviewer #2, Comment #4: For HIV infected patients, same day therapy seems to be helpful to reduce the incidence! According to your model, if we further shortening the duration of diagnosis to treatment of HCV, will it be more helpful to achieve the goal of elimination of HCV infection?

Author reply: Yes, we agree that our results support that further shortening the duration between time to diagnosis and treatment adds additional prevention benefits. As such, this strategy could be a useful tool to achieving HCV elimination. We further highlight this benefit in the Discussion section. 

Discussion, page 17, lines 298-300; 308-310: “However, reaching the HCV incidence elimination target is possible through the implementation of combination strategies which shorten the time between diagnosis and treatment and treat previously diagnosed and untreated. This underscores the importance of exploring strategies which do not solely rely on treatment alone but may require programmatic changes which has also been explored in other settings...”

---

## [Decision Letter · Decision Letter 1]

18 Apr 2022

Establishing a framework towards monitoring HCV microelimination among men who have sex with men living with HIV in Germany: A modeling analysis

PONE-D-22-03984R1

Dear Dr. Marquez,

We’re pleased to inform you that your manuscript has been judged scientifically suitable for publication and will be formally accepted for publication once it meets all outstanding technical requirements.

Kind regards,

Chen-Hua Liu

Academic Editor

PLOS ONE

Reviewers' comments:

Reviewer #1: All comments have been addressed

2. Is the manuscript technically sound, and do the data support the conclusions?

Reviewer #1: Yes

3. Has the statistical analysis been performed appropriately and rigorously? 

Reviewer #1: Yes

4. Have the authors made all data underlying the findings in their manuscript fully available?

Reviewer #1: Yes

5. Is the manuscript presented in an intelligible fashion and written in standard English?

Reviewer #1: Yes

6. Review Comments to the Author

Reviewer #1: I fully accept author's revision of manuscript and responses to my comments. Also, the author reply my comments with adequate references.

7. PLOS authors have the option to publish the peer review history of their article (what does this mean?). If published, this will include your full peer review and any attached files.

Reviewer #1: **Yes: **Cheng Chien-Yu

---

## [Editor Report · Acceptance letter]

3 May 2022

PONE-D-22-03984R1 

Establishing a framework towards monitoring HCV microelimination among men who have sex with men living with HIV in Germany: A modeling analysis 

Dear Dr. Marquez:

I'm pleased to inform you that your manuscript has been deemed suitable for publication in PLOS ONE. Congratulations! Your manuscript is now with our production department. 

Kind regards, 

on behalf of

Dr. Chen-Hua Liu 

Academic Editor

PLOS ONE